# Learning Fast and Slow: Addressing Task-Imbalanced Continual Learning with Decoupled Dual-Speed Adaptation

## Abstract

Continual learning aims to acquire new knowledge without forgetting previously learned tasks. However, most existing studies assume balanced tasks, which is rarely the practice case, as real-world scenarios often exhibit severe task imbalance with long-tailed distributions. This task-imbalanced continual learning (TICL) setting entangles two fundamental challenges: the well-known *stability–plasticity dilemma*, and the newly emerging *head–tail learning dilemma*, where head classes dominate training while tail classes remain under-optimized. To address this compounded difficulty, we propose Decoupled Fast–Slow Adaptation (DFSA), which introduces two key components. First, a fast–slow dual adapter augments the image encoder with a fast-adapting branch for rapid task acquisition and a slow-consolidating branch for stable knowledge retention. A task-modulated weighting mechanism dynamically integrates these branches, effectively fusing "fast" and "slow" thinking to balance short-term plasticity with long-term stability, while simultaneously providing complementary perspectives that enhance learning for underrepresented classes. Complementarily, DFSA employs a decoupled training strategy by first fine-tuning the text encoder as a semantic-aware classifier before updating image features, providing stable guidance that mitigates the negative impact of long-tailed distributions. Extensive experiments on TICL benchmarks show that our method significantly improves both few-sample task generalization and overall retention, outperforming existing continual learning baselines. The source code is temporarily available at https://anonymous.4open.science/r/DFSA-3aD6E.

## 1 Introduction

Human beings possess the remarkable ability to acquire new skills and adapt to evolving environments while retaining knowledge from past knowledge. Inspired by this, enabling deep models to achieve such continual learning (CL) (Wang et al., 2024a; Liu et al., 2025a) ability in artificial intelligence has been a long-standing goal, as it would allow models to operate robustly in dynamic and open-ended environments. Despite recent advances (Liu et al., 2025b; Wu et al., 2025; Huang et al., 2024; Liu et al., 2023; Thengane et al., 2022) driven by pre-trained foundation models such as CLIP (Radford et al., 2021), CL remains fundamentally constrained by the stability–plasticity dilemma: excessive plasticity causes catastrophic forgetting, while excessive stability impedes the acquisition of novel concepts.

The challenge becomes even more severe under real-world scenarios, where data typically follow a long-tailed distribution (Li et al., 2024; Zhang et al., 2023; Li et al., 2022) and arrive sequentially. In such cases, data streams often exhibit an extremely imbalanced (Hong et al., 2024; Liu et al., 2022): particular tasks contain abundant samples, whereas many others provide far fewer. To capture this more general and realistic setting, researchers have recently introduced task-imbalanced continual learning (TICL) (Hong et al., 2024). This imbalance introduces dual challenges, namely learning rare classes effectively while mitigating catastrophic forgetting. In essence, models in TICL are internally constrained by the stability–plasticity dilemma and externally challenged by the head–tail learning dilemma.

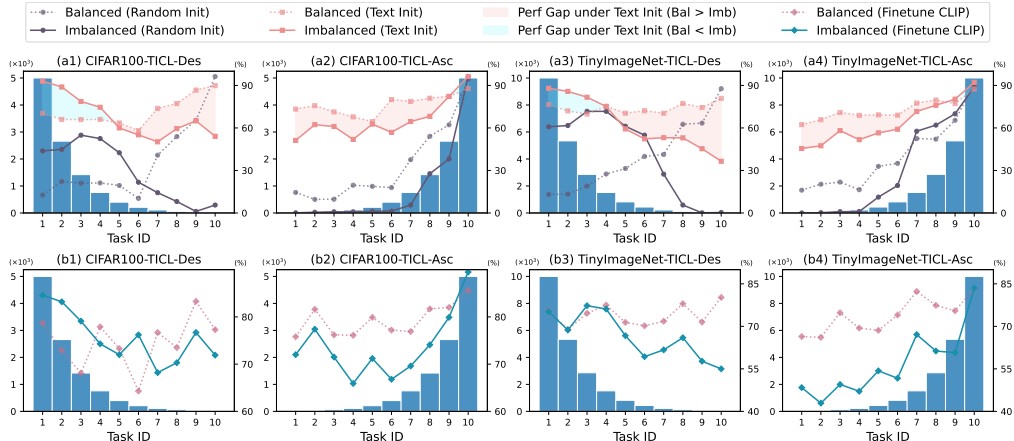

Figure 1: Final evaluation of the model on all learned tasks after sequential training. We conducted experiments on CIFAR100-TICL and TinyImageNet-TICL using classifiers (a1–a4) initialized either with text or randomly, as well as fine-tuning CLIP models (b1–b4).

Existing method joint training image representation and classifier (Hong et al., 2024; Liang & Li, 2024; Liu et al., 2023), or utilizing text guidance from the visual language models (Liu et al., 2025b; Yu et al., 2024; Zheng et al., 2023). However, training only the image encoder alongside a classifier often results in drastic performance drops on tail classes, as shown in Figure 1 (a1-a4), illustrating the intrinsic difficulty of learning under long-tailed distributions, and exhibiting patterns similar to those observed in regular long-tailed visual recognition (Shi et al., 2025; Li et al., 2023; Kang et al., 2020). More critically, certain tasks are almost completely forgotten across multiple classes (e.g., tasks 1–4 in Figure 1 a4). While methods incorporating the text encoder of visual-language models (VLMs) can alleviate part of the long-tail issue, they still struggle with pronounced imbalance. For example, in ascending distributions, early tail classes remain severely underrepresented and prone to catastrophic forgetting.

To address these joint challenges, we propose the Decoupled Fast–Slow Adaptation (DFSA) framework. Inspired by "Thinking, Fast and Slow" (Kahneman, 2011), DFSA introduces a Fast–Slow Adapter equipped with a task-modulated weighting mechanism that dynamically integrates a fast-adapting branch (analogous to Kahneman's System 1) for rapid task acquisition and a slow-consolidating branch (analogous to System 2) for stable knowledge retention. This adaptive fusion allows both "fast" and "slow" knowledge to synergistically guide learning, improving overall task adaptation while mitigating catastrophic forgetting. It also provides complementary perspectives for each task, enhancing representations of under-sampled classes and mitigating long-tailed bias. Complementarily, DFSA leverages the unique structure of visual–language models through decoupled training. The text encoder, acting as a semantic-aware classifier, is fine-tuned before the image encoder. Since text features converge faster (Ma et al., 2025; Hentschel et al., 2022; Radford et al., 2021), this step provides stable guidance that reduces the negative impact of long-tailed distributions on underrepresented classes.

Extensive experiments demonstrate the effectiveness of DFSA in TICL, showing that it can achieve nearly comparable balanced performance across tasks in continual learning. Our main contributions are summarized as follows:

- To address the stability–plasticity dilemma, we propose the Fast–Slow Adapter, equipping the image encoder with a fast branch for rapid task adaptation and a slow branch for long-term consolidation. A task-modulated weighting mechanism dynamically fuses these branches, allowing fast and slow knowledge to guide learning and mitigate catastrophic forgetting synergistically.

- To mitigate the head–tail learning dilemma, DFSA employs a unique decoupled training strategy that first fine-tunes the text encoder of VLMs before updating image features. The rapidly converging text encoder effectively serves as a classifier, providing stable semantic guidance to calibrate subsequent image feature learning.

- By integrating the Fast–Slow Adapter with decoupled text-encoder training, DFSA achieves near-balanced continual learning performance across tasks, demonstrating strong effectiveness under task-imbalanced long-tailed scenarios.

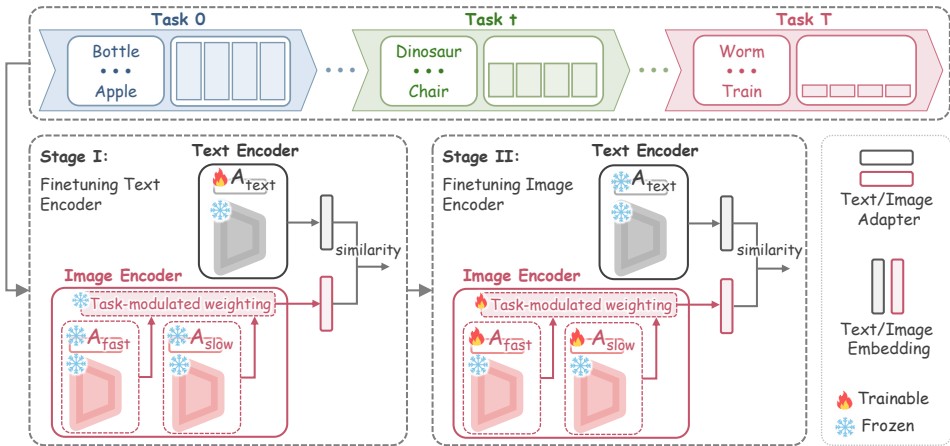

Figure 2: Training pipeline of DFSA, combining the Fast–Slow Adapter to alleviate the stability–plasticity dilemma and the decoupled training strategy to tackle the head–tail learning dilemma.

## 2 PROPOSED METHOD: DECOUPLED FAST–SLOW ADAPTATION

### 2.1 PRELIMINARIES

**Problem Definition.** We consider task-imbalanced continual learning (TICL), where the number of samples per task varies significantly throughout the learning process. In the conventional task-balanced continual learning setting with a sequence of $T+1$ tasks $\{\mathcal{D}_0, \mathcal{D}_1, \ldots, \mathcal{D}_T\}$, each task $\mathcal{D}_t$ contains $N_t$ samples of equal size, i.e., $N_i = N_j, \quad \forall i \neq j, \ i, j \in \{0, \ldots, T\}$. In contrast, TICL relaxes this constraint and considers the more realistic scenario where task sizes follow a non-uniform distribution. Motivated by the long-tailed nature of real-world data and the setup in Hong et al. (2024), we assume that task sizes also follow a long-tailed distribution. Formally, let $\{I_0, I_1, \ldots, I_T\}$ denote the task indices sorted in non-increasing order of size ($N_{I_0} > N_{I_1} > \cdots > N_{I_T}$). Depending on the arrival order of tasks, TICL can manifest in three variants: *Descending (long-tailed) order*, tasks arrive as $\{I_0, I_1, \ldots, I_T\}$, i.e., larger tasks first and smaller tasks later; *Ascending order*, tasks arrive in the reverse order $\{I_T, I_{T-1}, \ldots, I_0\}$, i.e., smaller tasks first and larger tasks later; *Random order*, tasks arrive in a random permutation, independent of their sizes.

**Parameter-Efficient Fine-Tuning (PEFT).** Instead of updating all parameters of large vision-language models, PEFT techniques introduce lightweight trainable modules while keeping the pretrained backbone frozen. Among them, LoRA (Hu et al., 2022), Adapter (Houlsby et al., 2019) and its variant AdaptFormer (Chen et al., 2022) are widely adopted. LoRA (Hu et al., 2022) constrains parameter updates to a low-rank subspace by decomposing $\Delta W \in \mathbb{R}^{m \times n}$ into $AB$ with $A \in \mathbb{R}^{m \times r}$ and $B \in \mathbb{R}^{r \times n}$, where $r \ll \min(m, n)$, so that the updated layer becomes $h' = (W_0 + AB)x$ with $W_0$ frozen. Adapters (Houlsby et al., 2019) insert small bottleneck modules into each transformer block, mapping $h \in \mathbb{R}^d$ through a down-projection, nonlinearity, and up-projection as $h' = h + W_{\text{up}} \, \sigma(W_{\text{down}} h)$. AdaptFormer (Chen et al., 2022) further stabilizes adapters by adding a LayerNorm and a learnable scaling factor $\alpha$, yielding $h' = h + \alpha \cdot W_{\text{up}} \, \sigma(W_{\text{down}} \, \text{LN}(h))$. In our framework, any of these can be employed to construct the Dual-Speed Adapter, allowing flexible and efficient adaptation.

### 2.2 DECOUPLED FAST–SLOW ADAPTATION

TICL presents two intertwined challenges: the stability–plasticity dilemma makes it difficult to retain past knowledge without sacrificing adaptation to new tasks, while the long-tailed distribution leads to biased representations that hurt underrepresented tasks. To address these challenges, we introduce the Fast–Slow Adapter, which combines a fast branch for rapid task adaptation with a slow branch for long-term knowledge consolidation. A task-modulated weighting mechanism dynamically fuses these branches, allowing their contributions to adapt per task. By ensuring that both fast-adapted and slow-consolidated knowledge are leveraged for each task, the model can better rep-

resent under-sampled (tail) classes, thereby mitigating the bias caused by long-tailed distributions while alleviating the stability–plasticity dilemma. To further reduce the impact of long-tailed distributions, we adopt a decoupled learning strategy by fine-tuning the text encoder before updating the image encoder, leveraging the faster convergence and semantic reliability of text features. Integrating these components, we propose the Decoupled Fast–Slow Adaptation (DFSA) framework for effective TICL. The pipeline of DFSA is illustrated in Figure 2.

### 2.2.1 DUAL SPEED ADAPTER

Motivated by Kahneman's dual-process theory (Kahneman, 2011), where "System 1" enables fast but short-lived adaptation and "System 2" supports slow but persistent consolidation, we design a dual-speed adaptation mechanism for the image encoder. In this analogy, the fast adapter plays the role of "System 1", capturing rapid but transient adjustments, while the slow adapter corresponds to "System 2", providing gradual and stable consolidation. Adapters are lightweight modules with only a small number of parameters (e.g., LoRA (Hu et al., 2022), Adapter (Houlsby et al., 2019), Adapt-Former (Chen et al., 2022)). We take the standard Adapter (Houlsby et al., 2019) as an example to illustrate the module design.

**Fast-Slow Adapter.** Building upon the standard adapter design, we extend it into two complementary branches within the image encoder: a fast adapter and a slow adapter. The fast adapter enables rapid adaptation to new tasks by capturing task-specific patterns while leaving the base encoder largely unchanged, whereas the slow adapter gradually consolidates knowledge to maintain long-term stability.

Given an input feature $h \in \mathbb{R}^d$, the standard adapter is formulated as

$$\text{Adapter}(h) = h + W_{\text{up}}\sigma(W_{\text{down}}h), \tag{1}$$

where $W_{\text{down}} \in \mathbb{R}^{r \times d}$, $W_{\text{up}} \in \mathbb{R}^{d \times r}$, and $\sigma(\cdot)$ denotes a non-linear activation. We extend this structure into two complementary branches with different adaptation speeds. The fast and slow branches are defined as

$$\text{A}_{\text{fast}}(h) = h + \alpha\, W_{\text{up}}^{\text{fast}} f(W_{\text{down}}^{\text{fast}}h), \quad \text{A}_{\text{slow}}(h) = h + \beta\, W_{\text{up}}^{\text{slow}} f(W_{\text{down}}^{\text{slow}}h), \tag{2}$$

where $\alpha > \beta$.

As shown in Figure 3, the accuracy of some tasks does not monotonically decrease. Instead, they occasionally improve after learning new tasks. This indicates that both past knowledge and current knowledge are beneficial, and their proper combination can lead to better performance. We introduce a task-modulated weighting strategy to dynamically balance the contributions of the fast and slow adapters. Given image features $F \in \mathbb{R}^{B \times d}$ for a batch of size $B$, we compute a task context as $c = \frac{1}{B}\sum_{i=1}^{B} F_i$, and obtain a modulating coefficient

$$\gamma = \sigma\left(\text{MLP}(c)\right) \in (0, 1), \tag{3}$$

where $\sigma(\cdot)$ is the sigmoid activation. Then, the final Fast–Slow Adapter is written as

$$\text{FSA}(h) = \gamma \cdot \text{A}_{\text{fast}}(h) + (1 - \gamma) \cdot \text{A}_{\text{slow}}(h). \tag{4}$$

The modulated coefficient $\gamma$ is learned jointly with the Fast-Slow Adapter in an end-to-end manner. Both adapters are lightweight, introducing only a small number of trainable parameters, which allows the image encoder to rapidly adapt to new tasks through the fast adapter. while maintaining long-term stability via the slow adapter.

### 2.2.2 DECOUPLED LEARNING OF TEXT AND IMAGE ENCODER.

**Empirical Observation.** In TICL, the impact of long-tailed distributions is further amplified over sequential updates. First, *head-class bias* arises because the prevalence of head-class samples induces head-class dominance, resulting in biased decision boundaries and severely compressed tail-class representations. Second, *cumulative bias* emerges as this imbalance accumulates across tasks: in settings such as ascending or random orders, the skew compounds over time, leading to increasingly distorted feature spaces and catastrophic forgetting. To mitigate such issues, decoupled learning methods (Zhong et al., 2021; Kang et al., 2020; Zhou et al., 2020) in long-tailed visual recognition typically adopt a two-stage strategy by learning a feature representation first, and then

re-balancing the classifier with balanced data sampling. However, this paradigm is suboptimal for continual learning. As image feature representations $f(x)$ are sequentially updated with new tasks, a classifier trained on previous features becomes misaligned with the current feature space, causing significant catastrophic forgetting.

To empirically examine this issue, we compare two decouple learning strategies: first fine-tuning the image encoder (Img Enc) versus first fine-tuning the text encoder (Text Enc). The results, presented in Figure 3, demonstrate clear differences in stability and performance. These findings motivate the following observation.

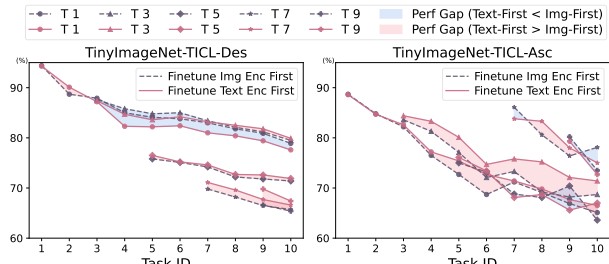

Figure 3: Accuracy evolution on 5 representative tasks (selected for clarity). Performance Gap (Perf Gap) shows text-first fine-tuning generally outperforms image-first, with occasional exceptions.

**Observation 1** (Feature Drift Amplifies Forgetting). *In continual learning, the sequential updates of image features $f(x)$ induce a mismatch between the classifier and the shifting feature space, causing decision boundary misalignment and consequently aggravating forgetting.*

This observation highlights the risk of feature drift. If the classifier is adjusted after feature updates, it will repeatedly fall out of alignment with the evolving representation space. To mitigate this issue, we adopt the opposite strategy. Instead of adapting the classifier to drifting features, we first fix a relatively balanced classifier that serves as a stable semantic reference. Subsequent image features are then encouraged to align with this reference, thereby reducing head-class dominance and alleviating forgetting. Building upon this intuition, we formalize the design principle in the following proposition.

**Proposition 1** (Fast-first margin stabilization). *Updating the fast-converging component before the slow component establishes a stable semantic anchor across classes. This stabilizes the class-wise margins and reduces the risk of bias accumulation, thereby alleviating class imbalance and mitigating forgetting during sequential updates.*

*Analysis.* Let $g$ denote the fast-converging component and $f$ the slow component. Let $\mu_c$ be the feature prototype of class $c$, i.e., the mean of $f(x)$ over class-$c$ samples.

When $g$ is trained to near convergence, each $g(c)$ serves as a stable semantic anchor for class $c$. Subsequent updates to $f$ adjust $\mu_c$ primarily to better align with $g(c)$. Because $g$ is discriminative across classes, these updates: (1) Move each $\mu_c$ toward its own anchor $g(c)$; (2) Limit excessive separation between head and tail classes; (3) Reduce feature drift across sequential tasks.

A simple first-order analysis at the class prototype level shows that the expected update direction for each class $c$ satisfies

$$\mathbb{E}[\Delta\mu_c] \propto g(c) - \mu_c + \epsilon_c,$$

where $\epsilon_c$ is a small residual due to inter-class interference. Since $g(c)$ is nearly fixed and discriminative, the residual is bounded and $\mu_c$ remains close to its anchor. Therefore, class-wise margins remain controlled, preventing uncontrolled drift or bias accumulation.

Fixing the fast-converging component first provides a reliable semantic reference. This guides the slow component to evolve its features in a stable and balanced manner, which reduces the risk of head-class dominance and mitigates forgetting during sequential updates. This analysis formalizes the intuition behind the fast-first design principle without requiring exact margin inequalities or heavy assumptions on tail classes. □

**Theorem 1** (Fast convergence of text encoder). *Under standard optimization settings, the text encoder converges faster than the image encoder.*

*Proof Sketch.* Text data are typically lower-dimensional and semantically more structured than image data (Ma et al., 2025; Radford et al., 2021; Yin et al., 2019). As a result, the corresponding encoder parameters reach a near-optimal solution in fewer gradient steps. Fixing this fast-converging

Table 1: Comparison Results (%) on ImageNet-R-TICL.

| Method | Descending | | Ascending | | Shuffled | |
|---|---|---|---|---|---|---|
| | $\overline{ACC}\uparrow$ | $ACC\uparrow$ | $\overline{ACC}\uparrow$ | $ACC\uparrow$ | $\overline{ACC}\uparrow$ | $ACC\uparrow$ |
| Pre+iCaRL (Rebuffi et al., 2017) | 48.41 | 29.55 | 24.40 | 29.17 | 40.21 | 23.02 |
| BiC (Wu et al., 2019) | 21.91 | 18.9 | 13.52 | 16.01 | 16.3 | 16.32 |
| PODNET (Douillard et al., 2020) | 22.32 | 18.90 | 16.61 | 16.62 | 17.11 | 18.70 |
| EEIL++ (Liu et al., 2022) | 18.50 | 17.81 | 15.76 | 15.60 | 15.79 | 16.30 |
| LUCIR++ (Liu et al., 2022) | 18.36 | 21.29 | 8.05 | 8.27 | 15.62 | 15.62 |
| Pre+FT (Khan et al., 2023) | 40.60 | 7.68 | 18.22 | 21.15 | 21.37 | 22.62 |
| Continual-CLIP (Thengane et al., 2022) | 81.62 | 74.27 | 77.86 | 74.30 | 80.59 | 74.23 |
| L2P (Wang et al., 2022b) | 69.19 | 55.83 | 39.59 | 53.73 | 55.80 | 55.15 |
| DualPrompt (Wang et al., 2022a) | 65.99 | 51.03 | 38.03 | 51.43 | 50.01 | 50.65 |
| CODA-Prompt (Smith et al., 2023) | 73.98 | 60.72 | 42.33 | 59.15 | 58.19 | 58.37 |
| DAP (Hong et al., 2024) | 57.19 | 42.62 | 36.39 | 44.80 | 47.11 | 44.53 |
| MoE-Adapters (Yu et al., 2024) | 86.50 | 79.80 | 78.68 | 79.23 | 83.83 | 79.15 |
| LoRA-DRS (Liu & Chang, 2025) | 78.83 | 66.90 | 38.61 | 55.08 | 56.13 | 60.38 |
| SD-LoRA (Wu et al., 2025) | 77.29 | 66.75 | 50.67 | 65.07 | 65.86 | 66.18 |
| DFSA (Ours) | **87.66** | **81.37** | **80.78** | **79.68** | **84.58** | **80.67** |

component establishes stable class-wise embeddings, which then guide the slower-evolving image encoder to align its features accordingly. This reduces the risk of misalignment and mitigates forgetting without requiring explicit rebalancing at every step. □

Detailed analysis is provided in Appendix A. According Theorem 1 and Proposition 1, fine-tuning the text encoder first provides a reliable semantic anchor, guiding the subsequent learning of image features and mitigating feature drift across sequential tasks.

## 3 EXPERIMENTS

### 3.1 EXPERIMENTAL SETUP

**Datasets and Evaluation Metrics.** We evaluate our approach on three widely used benchmarks: CIFAR100-TICL, TinyImageNet-TICL, and ImageNet-R-TICL, which are derived from the original CIFAR100 (Krizhevsky et al., 2009), TinyImageNet (Le & Yang, 2015), and ImageNet-R (Hendrycks et al., 2021), respectively. CIFAR100 contains 100 classes with 500 images per class, while TinyImageNet contains 200 classes with 500 images per class sampled from ImageNet (Russakovsky et al., 2015). ImageNet-R includes 200 classes rendered in diverse artistic styles such as cartoons, graffiti, and origami, providing a challenging out-of-distribution variant of ImageNet.

Following the TICL protocol introduced by Hong et al. (2024), each dataset is restructured into a long-tailed variant and further divided into 10 disjoint tasks, forming their TICL counterparts: CIFAR100-TICL, TinyImageNet-TICL, and ImageNet-R-TICL. The data distribution of TICL is shown in Appendix B. TICL simulates more realistic incremental learning under long-tailed data distributions by considering three task-arrival scenarios: (1) *TICL-Descending*, where tasks arrive from head-class rich (data-abundant) to tail-class dominated (data-scarce); (2) *TICL-Ascending*, where tasks arrive in the reverse order, i.e., from tail to head; (3) *TICL-Shuffled*, where tasks arrive in a random sequence.

To assess model performance, we adopt two standard continual learning metrics: (1) *Average accuracy* ($\overline{ACC}$), which averages the accuracy over all encountered tasks at each training step, providing a more fine-grained measure of continual performance throughout learning; and (2) *Accuracy after the last stage* ($ACC$), which measures the accuracy on all tasks after the final task is learned, reflecting the overall effectiveness of sequential training. More specifically, let $ACC_t$ denote the accuracy of the model after completing the $t$-th task. $\overline{ACC} = \frac{1}{T}\sum_{t=1}^{T} ACC_t$, and $ACC = ACC_T$.

**State-of-the-art Methods.** We compare the proposed DFSA against state-of-the-art approaches based on fine-tuning pre-trained models. (1) *Rehearsal-based methods*: PODNET (Douillard et al.,

Table 2: Comparison Results (%) on CIFAR100-TICL.

| Method | Descending | | Ascending | | Shuffled | |
|---|---|---|---|---|---|---|
| | $\overline{ACC}$ ↑ | $ACC$ ↑ | $\overline{ACC}$ ↑ | $ACC$ ↑ | $\overline{ACC}$ ↑ | $ACC$ ↑ |
| Pre+iCaRL (Rebuffi et al., 2017) | 53.00 | 28.73 | 41.70 | 26.88 | 48.62 | 31.02 |
| BiC (Wu et al., 2019) | 27.92 | 38.02 | 36.08 | 37.90 | 27.11 | 33.63 |
| PODNET (Douillard et al., 2020) | 26.48 | 23.82 | 32.31 | 28.24 | 28.49 | 26.21 |
| EEIL++ (Liu et al., 2022) | 31.49 | 38.24 | 35.93 | 37.85 | 31.63 | 39.31 |
| LUCIR++ (Liu et al., 2022) | 27.71 | 21.26 | 42.39 | 28.92 | 35.62 | 25.94 |
| Pre+FT (Khan et al., 2023) | 65.83 | 22.02 | 19.52 | 25.58 | 43.30 | 33.56 |
| Continual-CLIP (Thengane et al., 2022) | 76.35 | 66.69 | 75.88 | 66.62 | 76.52 | 66.64 |
| L2P (Wang et al., 2022b) | 79.43 | 57.41 | 47.22 | 50.61 | 59.13 | 53.91 |
| DualPrompt (Wang et al., 2022a) | 81.76 | 63.21 | 54.96 | 50.96 | 65.22 | 57.21 |
| CODA-Prompt (Smith et al., 2023) | 78.09 | 51.65 | 58.20 | 53.19 | 69.08 | 61.58 |
| DAP (Hong et al., 2024) | 81.15 | 65.48 | 50.29 | 46.43 | 56.32 | 55.27 |
| MoE-Adapters (Yu et al., 2024) | 84.05 | 75.53 | 78.59 | 74.61 | 82.12 | 74.11 |
| DFSA (Ours) | **85.43** | **76.91** | **81.28** | **75.96** | **83.57** | **75.65** |
| LoRA-DRS[1] (Liu & Chang, 2025) | 91.69 | 83.37 | 75.99 | 76.21 | 84.84 | 78.02 |
| SD-LoRA[1] (Wu et al., 2025) | 88.42 | 76.64 | 73.20 | 72.34 | 81.41 | 72.93 |

Table 3: Comparison Results (%) on TinyImageNet-TICL.

| Method | Descending | | Ascending | | Shuffled | |
|---|---|---|---|---|---|---|
| | $\overline{ACC}$ ↑ | $ACC$ ↑ | $\overline{ACC}$ ↑ | $ACC$ ↑ | $\overline{ACC}$ ↑ | $ACC$ ↑ |
| L2P (Wang et al., 2022b) | 82.98 | 64.45 | 60.14 | 59.56 | 66.28 | 61.34 |
| DualPrompt (Wang et al., 2022a) | 81.53 | 60.17 | 58.16 | 61.71 | 66.31 | 60.82 |
| CODA-Prompt (Smith et al., 2023) | 80.79 | 59.28 | 64.81 | 61.14 | 71.98 | 61.74 |
| DAP (Hong et al., 2024) | 80.24 | 61.95 | 54.27 | 58.92 | 65.78 | 59.91 |
| Continual-CLIP (Thengane et al., 2022) | 74.69 | 65.92 | 73.39 | 65.94 | 74.38 | 65.95 |
| MoE-Adapters (Yu et al., 2024) | 81.75 | 72.92 | 76.31 | 72.29 | 78.88 | 72.24 |
| DFSA (Ours) | **83.24** | **74.12** | **77.47** | **73.16** | **80.11** | **73.48** |
| LoRA-DRS[1] (Liu & Chang, 2025) | 84.64 | 72.54 | 67.40 | 51.09 | 75.43 | 64.01 |
| SD-LoRA[1] (Wu et al., 2025) | 90.77 | 81.16 | 81.00 | 78.83 | 85.44 | 79.98 |

2020), and BiC (Wu et al., 2019), which are classical continual learning baselines. (2) *Long-tailed extensions*: EEIF++ (Liu et al., 2022) and LUCIR++ (Liu et al., 2022), designed to address class imbalance in continual learning. (3) *ViT-based methods*: L2P (Wang et al., 2022b), DualPrompt (Wang et al., 2022a), CODA-Prompt (Smith et al., 2023), FineTune (Khan et al., 2023), and DAP (Hong et al., 2024), which exploit prompt tuning (Jia et al., 2022) for task adaptation, as well as LoRA-based methods such as LoRA-DRS (Liu & Chang, 2025) and SD-LoRA (Wu et al., 2025). (4) CLIP-based methods: Continual-CLIP (Thengane et al., 2022), which leverages CLIP's zero-shot generalization in TICL, and MoE-Adapters (Yu et al., 2024), which employ mixture-of-experts adapters for continual learning. (5) Other pre-trained baselines: FineTune (Khan et al., 2023) and iCaRL (Rebuffi et al., 2017) with pre-trained ViT.

**Implementation Details.** As a widely adopted representative of VLMs (Yu et al., 2024; Huang et al., 2024), we exploit CLIP as our backbone, specifically the ViT-B/16 version. We fine-tune both its image and text encoders using Adapter (Houlsby et al., 2019) as the PEFT technology. For all datasets, the adaptation rates mentioned in Section 2.2.1 are set as $\alpha = 0.5$ for the fast branch and $\beta = 0.1$ for the slow branch.

## 3.2 COMPARISON RESULTS

Tables 1 to 3 present the results of different methods on ImageNet-R-TICL, CIFAR100-TICL and TinyImageNet-TICL. The comparison results demonstrate the consistent superiority of DFSA over

---

[1]The pre-training dataset is ImageNet-21K (Russakovsky et al., 2015; Ridnik et al., 2021), which overlaps with CIFAR100-TICL and TinyImageNet-TICL, potentially causing data leakage (Shi et al., 2024). Results on these benchmarks are therefore reported for reference only.

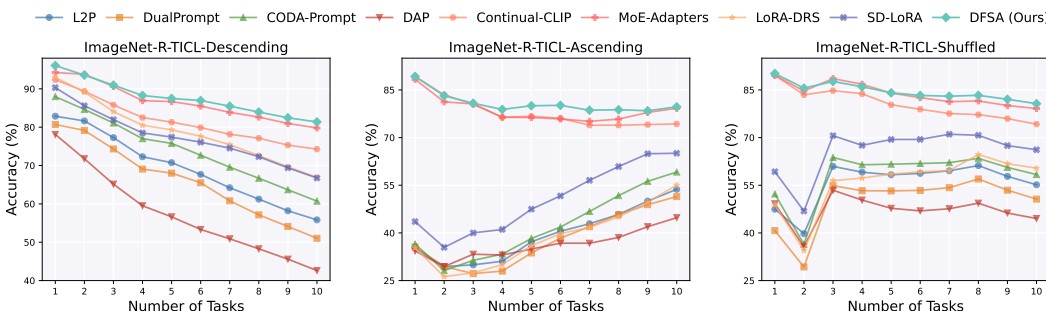

Figure 4: Average accuracy on all encountered tasks after sequential training on ImageNet-R-TICL.

Table 4: Ablation Results (%) on TinyImageNet-TICL. "Image-Text" updates the image encoder before the text encoder, while "Text-Image" indicates the reverse.

| Decoupled Learning | | FSA | Descending | | Ascending | | Shuffled | |
|---|---|---|---|---|---|---|---|---|
| Image-Text | Text-Image | | $\overline{ACC}$ ↑ | $ACC$ ↑ | $\overline{ACC}$ ↑ | $ACC$ ↑ | $\overline{ACC}$ ↑ | $ACC$ ↑ |
| ✓ | ✗ | ✗ | 81.69 | 72.51 | 76.72 | 70.85 | 79.17 | 72.09 |
| ✗ | ✓ | ✗ | 82.52 | 73.08 | 77.01 | 72.64 | 79.85 | 72.37 |
| ✓ | ✗ | ✓ | 82.53 | 73.62 | 77.25 | 72.00 | 79.02 | 72.40 |
| ✗ | ✓ | ✓ | **83.24** | **74.12** | **77.47** | **73.16** | **80.11** | **73.48** |

state-of-the-art baselines. On ImageNet-R-TICL, DFSA achieves substantial improvements, surpassing recent strong baselines by over 10% in both $\overline{ACC}$ and $ACC$. Notably, compared to SD-LoRA (with a ViT-B/16-IN21K backbone), DFSA achieves remarkable gains under the Ascending order (+30.11% $\overline{ACC}$ and +14.61% $ACC$), showing its strong adaptability and stability. Against MoE-Adapters (a CLIP-based method), DFSA also delivers steady improvements, such as a +2.1% gain in $\overline{ACC}$ under Ascending. On CIFAR100-TICL and TinyImageNet-TICL, DFSA further outperforms all compared methods (except methods with ViT-B/16-IN21K backbone) across all task orders. In particular, on CIFAR100-TICL, DFSA improves upon MoE-Adapters by more than 1.3% in every scenario, with $\overline{ACC}$ gains reaching +2.69% under Ascending order. Consistent gains are also observed on TinyImageNet-TICL. Although SD-LoRA and LoRA-DRS are excluded from the comparisons due to pretraining overlap with ImageNet-21K, DFSA still surpasses them in several cases. For instance, on CIFAR100-TICL Ascending, DFSA outperforms LoRA-DRS and SD-LoRA by +5.29% and +8.08% in $\overline{ACC}$, respectively.

Figure 4 shows the sequential performance of various methods on ImageNet-R-TICL. In the Descending scenario, where head tasks arrive first, all methods start strong but gradually decline due to forgetting and underrepresented of tail tasks. DFSA declines the slowest, showing its superior robustness against forgetting. In the Ascending scenario, initial tasks have few samples, causing most methods to perform poorly initially. The arrival of larger tasks with richer samples leads to higher performance on these tasks, thereby raising the average accuracy. DFSA effectively leverages textual guidance to maintain strong performance throughout. Compared with MoE-Adapters, which also use textual information, DFSA exhibits a smoother accuracy curve, demonstrating its enhanced ability to mitigate catastrophic forgetting under extreme data scarcity. Additional results on CIFAR100-TICL and TinyImageNet-TICL are provided in Appendix C. Appendix D presents a comparison of per-task performance after training the final task.

### 3.3 ABLATION STUDY

Table 4 presents the ablation results evaluating the contributions of individual DFSA components. When decoupling the learning of text and image encoders, the Text-Image order consistently outperforms Image-Text, with a notable +1.79% $ACC$ gain under the Ascending scenario, demonstrating that stabilizing the decision boundary before feature refinement is crucial for long-tailed tasks. Adding the Fast-Slow Adapter further boosts performance. Image-Text can also benefit from the Fast-Slow Adapter (FSA) in both Descending and Ascending, while in Shuffled it incurs only a marginal drop of 0.15% in $\overline{ACC}$. Finally, combining Text-Image with the FSA yields the best results across all scenarios, underscoring the complementary roles of both components in DFSA.

## 4 RELATED WORK

**Continual Learning (CL).** Existing CL approaches primarily seek to mitigate catastrophic forgetting under the assumption of balanced task distributions and can broadly be categorized into three main types (Wang et al., 2024a; De Lange et al., 2021): architecture-based, regularization-based, and rehearsal-based methods. Architecture-based methods (Mallya et al., 2018; Rypeść et al., 2023; Li et al., 2025) adjust the model architecture and allocate specific parameters for different tasks. Regularization-based methods (Dhar et al., 2019; Lee et al., 2019; Zenke et al., 2017; Kirkpatrick et al., 2017) restrict parameter optimization by knowledge distillation or adding a penalty in the loss function, thus accommodating both previous and new tasks. Rehearsal-based methods (Rebuffi et al., 2017; Rolnick et al., 2019; Kumari et al., 2022) mitigate forgetting by replaying original or synthesized samples from previous tasks, which typically results in additional memory overhead and potential privacy risks.

**Fine-tuning Pre-trained Model for CL.** Recently, fine-tuning pre-trained models using parameter-efficient fine-tuning (PEFT) methods (Jia et al., 2022; Chen et al., 2022; Hu et al., 2022) has shown strong performance in CL. Some CL methods, such as L2P (Wang et al., 2022b), DualPrompt (Wang et al., 2022a) and CODA-Prompt (Smith et al., 2023) combined pre-trained Vision Transformer (ViT) (Dosovitskiy et al., 2021) with prompt tuning (Lester et al., 2021; Li & Liang, 2021). Additionally, several methods, like InfLoRA (Liang & Li, 2024) and SD-LoRA (Wu et al., 2025), fine-tune pre-trained ViT using LoRA (Hu et al., 2022) for CL. CLIP (Radford et al., 2021) contains knowledge from both the language and vision modalities and exhibits strong CL performance under zero-shot evaluation (Thengane et al., 2022). Liu et al. (2023) fine-tunes the image encoder using an adapter and preserves previous knowledge through a parameter retention mechanism and knowledge distillation. RAPF (Huang et al., 2024) also employs an adapter to adjust image features with a decomposed parameter fusion strategy to enhance stability, and utilizes textual information to separate neighboring categories. Yu et al. (2024) employs mixture-of-experts Adapters to fine-tune the image and text encoders of CLIP, and integrates the outputs of experts based on the input sample.

**Imbalanced Continual Learning.** This setting is characterized by two intertwined challenges: catastrophic forgetting and data imbalance, which exacerbate each other. Earlier research, such as BiC (Wu et al., 2019) and PRS (Kim et al., 2020), focuses on imbalanced continual learning based on sample rehearsal. Liu et al. (2022) defined long-tailed class-incremental learning (LT-CIL) and proposed a rehearsal-based two-stage approach Some studies (He, 2024; Wang et al., 2024b) mitigated the impact of data imbalance under LT-CIL (Liu et al., 2022). Recently, Hong et al. identified the challenges posed by task-level sample imbalance and introduced Task Imbalanced Continual Learning (TICL). In TICL, the sample sizes across tasks follow a long-tailed distribution, encompassing three scenarios: Descending, Ascending, and Shuffled TICL. DAP (Hong et al., 2024), a rehearsal-free two-stage method, learned task-specific prompts in the first stage, and then integrated them into a general prompt in the second stage according to task sizes. While this method effectively tackled the challenges in TICL, its performance still remains inadequate. This paper focuses on task-level imbalance and aims to further improve the performance of three scenarios of TICL.

## 5 CONCLUSION

In this work, we proposed the Decoupled Fast–Slow Adaptation (DFSA) framework to address the compounded challenges of task-imbalanced continual learning. By combining a Fast–Slow Adapter with a task-modulated weighting mechanism, DFSA effectively balances short-term plasticity and long-term stability, while also providing complementary perspectives that benefit underrepresented tail classes. In addition, the decoupled training strategy leverages the semantic guidance of text encoders to further reduce the adverse effects of long-tailed distributions. Extensive experiments across multiple benchmarks demonstrate that DFSA consistently outperforms state-of-the-art methods in both generalization to few-sample tasks and long-term knowledge retention.

Although DFSA demonstrates strong performance in TICL, its fast and slow adaptation speeds are manually set, which may not be optimal for all tasks. Future work will explore an automatic learning strategy for these adaptation speeds to better handle heterogeneous tasks. Moreover, while TICL settings better reflect real-world scenarios than traditional task-balanced assumptions, more challenging conditions, such as highly non-stationary or cross-domain distributions, remain under-explored. Extending DFSA to these settings could further improve robustness and generalization in practical continual learning.

## ETHICS STATEMENT

Our research focuses on effectively alleviating the plasticity and stability dilemma and the newly head–tail learning problem in TICL with the proposed DFSA, more applicable to real-world scenarios where task streams arrive continuously and task sizes follow long-tailed distributions. We have carefully considered the potential social impacts and anticipate no direct, immediate, or adverse consequences. We are committed to the ethical dissemination of our research and to encouraging its responsible use.

## REPRODUCIBILITY STATEMENT

All the results in this work are reproducible. We provide a temporary code in the supplemental materials and an anonymous repository linked in the Abstract to replicate our results. The repository includes environment configurations, run scripts, and other relevant materials. A complete version of the code will be released at a later stage. We discuss the experimental settings in Section 3.1, including implementation details such as the backbone and the configuration of PEFT technology.

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

APPENDIX: LEARNING FAST AND SLOW: ADDRESSING TASK-IMBALANCED CONTINUAL LEARNING WITH DECOUPLED DUAL-SPEED ADAPTATION

## A  DETAIL PROOF OF THEOREM 1

*Proof.* Let $g_\theta$ denote the text encoder and $f_\phi$ the image encoder. Denote the loss function over a task by $\mathcal{L}(f_\phi, g_\theta)$.

**Optimization landscape.** Text inputs are typically lower-dimensional, more structured, and carry strong semantic correlations. Image inputs are high-dimensional and more complex, with diverse patterns and noise. Thus, the Hessian $\nabla_\theta^2 \mathcal{L}$ for the text encoder tends to have larger eigenvalues along informative directions, leading to faster gradient descent convergence.

**Convergence rate comparison.** Using standard gradient descent with learning rate $\eta$:
$$\theta_{t+1} = \theta_t - \eta \nabla_\theta \mathcal{L}(\theta_t, \phi_t), \quad \phi_{t+1} = \phi_t - \eta \nabla_\phi \mathcal{L}(\theta_t, \phi_t),$$
the text encoder updates $\theta$ converge faster because:

1. Smaller input dimension $\Rightarrow$ fewer parameters and smoother loss landscape;
2. Semantically structured inputs $\Rightarrow$ gradients are more informative and less noisy;
3. Fewer conflicting updates across classes.

**Implication for continual learning.** Once $g_\theta$ (text encoder) is near convergence, it serves as a stable semantic anchor. Subsequent updates of $f_\phi$ (image encoder) are guided to align with $g_\theta$, stabilizing class-wise representations and mitigating feature drift and forgetting. $\square$

## B  VISUALIZATION OF DATA DISTRIBUTION UNDER TICL

Figure A1 depicts the data distribution under the TICL setting (Hong et al., 2024), exemplified by CIFAR100-TICL. The task sizes follow a long-tailed distribution, whereas the categories within each task are balanced.

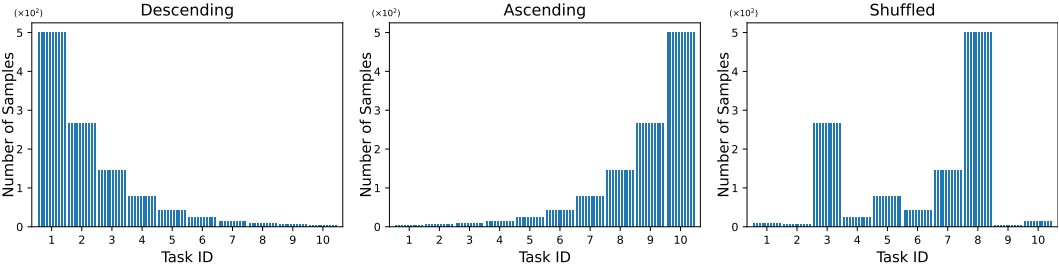

Figure A1: Illustration of the data distribution in CIFAR100-TICL. (The categories within each task can be clearly visible upon magnification.)

## C  SEQUENTIAL TASK PERFORMANCE ON CIFAR100-TICL AND TINYIMAGENET-TICL

Figure A2 (in Page 15) compares the average accuracy across all encountered tasks. DFSA consistently demonstrates robust performance. Notably, even when compared against methods pretrained on ImageNet-21K, which has overlap with the training data, DFSA outperforms some of these methods in the more challenging Ascending scenario (Figure A2a), showing its effectiveness under extreme task imbalance and limited early data.

## D  PER-TASK PERFORMANCE COMPARISON

Figure A3a (in Page 16) shows the per-task accuracy after learning the final task for methods using linear classifiers. It can be seen that task performance is highly sensitive to data volume, with

data-scarce tasks exhibiting extremely poor accuracy. In contrast, our DFSA maintains consistently strong performance across all tasks, demonstrating its stability in task-imbalanced continual learning and its effectiveness in mitigating both the stability–plasticity and head–tail learning dilemmas. Similarly, Figure A3b compares DFSA with textual classifier methods, namely Continual-CLIP and MoE-Adapters. While all methods leverage textual information to reduce sensitivity to data scarcity, DFSA achieves higher accuracy on more tasks, further showing its robustness under long-tailed incremental scenarios.

## DISCLOSURE OF LLM USAGE

In accordance with the ICLR 2026 policy, we disclose that large language models (e.g., ChatGPT, Deepseek) were used only to aid in the polishing of grammar and phrasing. Their usage was strictly limited to improving readability and presentation. They were not used for research ideation, methodology design, algorithm development, data collection, data analysis, experimental design, result interpretation, or any aspect related to the originality and technical contributions. All scientific contributions are original to the authors.

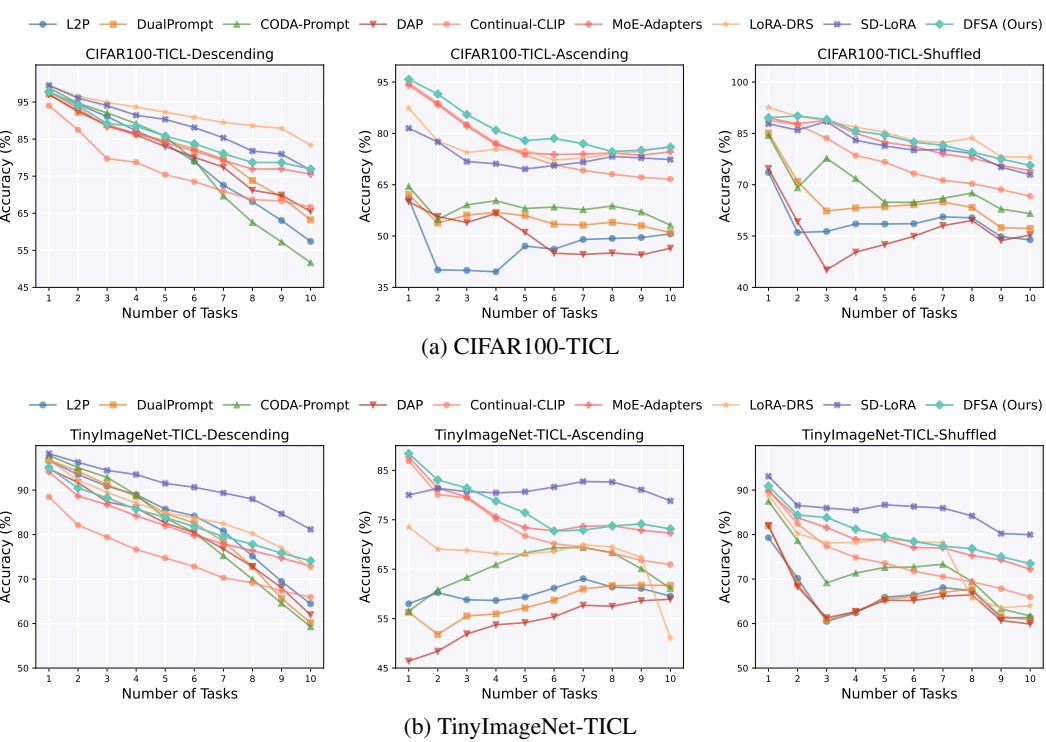

Figure A2: Average accuracy on all encountered tasks after sequential training on (a) CIFAR100-TICL, and (b) TinyImageNet-TICL.

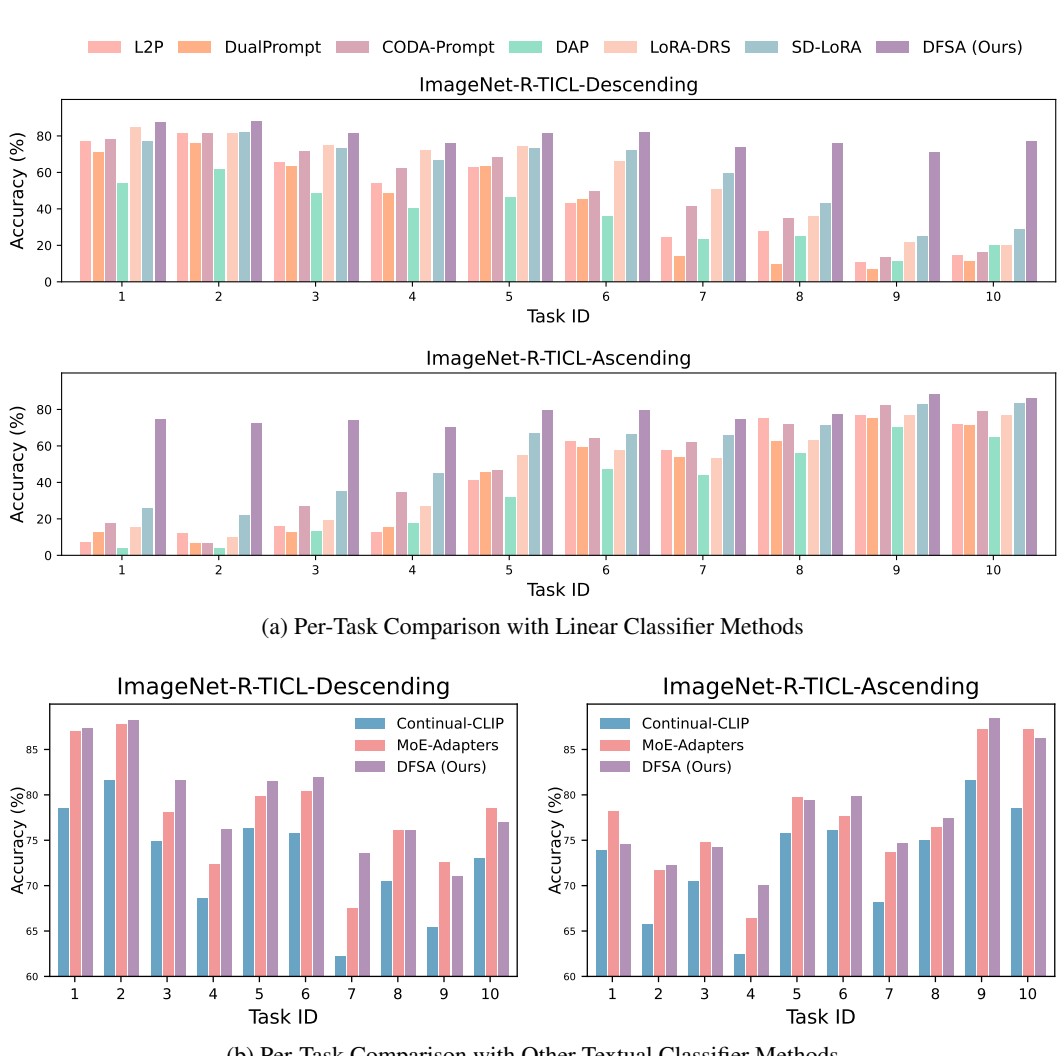

(a) Per-Task Comparison with Linear Classifier Methods

(b) Per-Task Comparison with Other Textual Classifier Methods

Figure A3: Per-task performance comparison after training final task: (a) our DFSA vs. linear classifier methods, and (b) our DFSA vs. textual classifier methods.

