# OpenReview forum: "Learning Fast and Slow: Addressing Task-Imbalanced Continual Learning with Dual-Speed Adaptation"
_ICLR.cc/2026/Conference — ICLR 2026 Conference Withdrawn Submission_

### Official Review · Reviewer_6Wi6 · 2025-10-29

**Soundness:** 3
**Presentation:** 3
**Contribution:** 3
**Rating:** 6
**Confidence:** 4

**Summary:**

This paper addresses Task-Imbalanced Continual Learning (TICL), a challenging setting combining the stability-plasticity dilemma of Continual Learning (CL) with the head-tail learning dilemma from long-tailed task distributions. It proposes Decoupled Fast-Slow Adaptation (DFSA), which integrates two core components: 1) A Fast-Slow Adapter (FSA) added to the image encoder, featuring parallel fast-adapting and slow-consolidating branches, dynamically fused via task-modulated weighting to balance plasticity and stability. 2) A Decoupled Training Strategy that first fine-tunes the faster-converging text encoder of a Vision-Language Model (VLM) to establish a stable semantic anchor, before subsequently adapting the image encoder features to align with it. DFSA demonstrates strong empirical performance, particularly excelling in the difficult Ascending TICL scenario.

**Strengths:**

[S1] Tackles the realistic and difficult TICL problem, concurrently addressing forgetting and long-tail bias.

[S2] Consistently outperforms state-of-the-art methods across TICL benchmarks, especially showing significant gains in the challenging Ascending task order, indicating effectiveness against both dilemmas.

[S3] Clearly validates the contribution of both the FSA and the specific "Text-Image" decoupled training order.

**Weaknesses:**

[W1] While FSA intuitively addresses stability-plasticity, its specific mechanism for alleviating the head-tail dilemma (enhancing tail class representation) lacks detailed explanation or analysis.

[W2] Sensitivity analysis for the fast ($\alpha$) and slow ($\beta$) adaptation rates is absent. Additionally, including an analysis of the trajectory of $\gamma across tasks would considerably enhance the interpretability and empirical soundness of the proposed dual-speed adaptation.

[W3] The weighting factor $\gamma$ appears computed based on batch features, not explicit task identity, making the term "task-modulated" potentially misleading.

[W4] Acknowledged data overlap between the backbone's pretraining (ImageNet-21K) and two benchmark datasets (CIFAR100, TinyImageNet) somewhat weakens the reported results on those specific datasets.

**Questions:**

Please refer to the weaknesses.

---

### Official Review · Reviewer_B4Ha · 2025-10-30

**Soundness:** 2
**Presentation:** 2
**Contribution:** 2
**Rating:** 2
**Confidence:** 4

**Summary:**

This paper proposes two techniques to improve the performance of TICL (task-imbalanced continual learning), in which data follows a long-tailed distribution across different tasks. TICL is originally proposed by Hong et al., (2024), and this work attempts to boost the performance over existing CL methods when applying to TICL. One component of this work is DFSA (decoupled fast-slow adaptation) that augments the image encoder with dual branches with different learning speeds, and the other is another decoupled learning strategy that fine-tunes the text encoder of a VLM before updating the image encoder. Using TICL variants of standard benchmarks (e.g., CIFAR100 and TinyImageNet, the proposed method shows improvement over existing baselines.

**Strengths:**

1. This work deals with a more realistic and challenging CIL setting, which was originally introduced by Hong et al., (2024)
2. The decoupled training strategy between text encoder and image encoder is justified by observation, proposition, and theorem.
3. The paper provides extensive comparisons with many existing baseline methods.

**Weaknesses:**

1. The paper is not well-written and lacks clarity in problem formulation. It is unclear what the exact goal of the problem, especially given that the method is built on a vision–language model (CLIP). The paper does not specify whether it performs vision-only classification or cross-modal alignment, which makes the problem definition incomplete. This becomes clear only after reading the experimental section.

2. The dual-speed mechanism lacks quantitative evidence. The difference between the fast and slow branches is manually fixed (\alpha=0.5, \beta=0.1) without any sensitivity analysis or ablation. There is no justification for why this ratio is optimal or even effective. The claimed “dual-speed” behaviour is therefore not empirically validated.

3. The learnable gating parameter \gamma may cancel the effect of \alpha and \beta. Since \gamma is end-to-end learnable, it can easily dominate the weighting between fast and slow adapters. In that case, the predefined ratio of \alpha and \beta becomes meaningless. The paper does not analyze the behaviour of \gamma or show that it actually preserves two distinct learning dynamics.

4. Theorem 1 is not a real theoretical proof, but heuristic and qualitative. It lacks any formal assumptions, mathematical derivation, or convergence bound. It functions more as an intuitive explanation than a rigorous theoretical result.

5. The data leakage argument is overstated. The authors state that competing methods trained on ImageNet-21K as “for reference only,” citing possible overlap with CIFAR100 or TinyImageNet. However, these datasets do not share actual images, only similar class names. The argument seems defensive rather than technically justified.

6. The originality is limited. The method largely follows the current trend of CLIP-based continual learning (e.g., Continual-CLIP, RAPF, MoE-Adapters). The proposed fast–slow learning strategy is a well-known paradigm in continual learning (e.g., Learning Fast, Learning Slow: A General Continual Learning Method based on Complementary Learning System, ICLR 2022).

**Questions:**

Pleases refer to each of the weaknesses above.

---

### Official Review · Reviewer_jvrY · 2025-10-30

**Soundness:** 2
**Presentation:** 3
**Contribution:** 2
**Rating:** 4
**Confidence:** 3

**Summary:**

This paper proposes an approach to imbalanced class incremental learning, where the overall distribution of classes to learn in a sequential manner is long tailed. Authors propose to design an approach that balances between fast adaptation/learning of task and slow consolidation of knowledge over time. To do this, they propose to use vision language models (e.g. CLIP) and decouple the fine-tuning procedure by adapting the text and image encoder sequentially. The fast-slow adaptation procedure leverages two separate adapters with a learnable task specific weight adjusting the relative influence of each adapter. The method is tested on multiple continual learning classification benchmarks designed for task imbalance, showing strong performance overall.

**Strengths:**

The idea of balancing fast adaptation to knowledge with slow consolidation of past knowledge is well justified and makes sense, in particular in the context of continual learning. Exploring the situation where classes are imbalanced is more realistic, and it’s great to see efforts to tackle more complex problems.

Relying on vision/language models to handle task imbalance and low data regimes for tail classes is sensible, as it generally allows more robust predictions. This is evidenced by experiments in Figure A3 where vision-language methods dominate for tail classes. It is nice to see authors make efforts to analyse the behaviour of models under different conditions, notably the impact of text models, to inform and motivate their methodology and ideas.

Experiments are thorough, covering multiple standard benchmarks and achieving strong performance overall.

**Weaknesses:**

While I like the idea of fast and slow learning, my main concern is the implementation of this idea via the dual adapter, which does not seem to reflect the fast adaptation and consolidation of knowledge discussed in prior sections. One potential reason for this is the limited intuition provided when introducing the dual adapters (sec 2.2.1). It is unclear how two identical adapters with learnable relative weighting (obtained from the average feature representation of a batch) can lead to a fast and slow learning procedure, especially as learned weights are never visualised or analysed. Wouldn’t an exponential moving average (EMA) based adapter be a more accurate implementation of the fast/slow idea?

It would be great to get more intuition and explanations on why this formulation enables slow consolidation of knowledge.
The sequential adaptation idea is interesting, and discussed in much more details. However, there is a lack of clarity in this section, and there are limited experiments validating this choice. Authors investigate the impact of the order of this sequential adaptation, but do not show the benefit of sequential adaptation vs adapting both, or keeping one fixed.

One important detail to clarify as well is to confirm that sequential adaptation is carried out per task, instead of first adapting the text encoder on all tasks, then adapting the visual encoder. The latter would violate the continual learning constraint, therefore I am assuming the former is used. However, it is unclear how adapting the classifier then the features per task reduces drifting issues across tasks. It would be great for authors to clarify their training protocol and how this classifier stability is achieved in a continual learning setting.

**Questions:**

-	As discussed in the weaknesses sections, it would be great to get intuition of why section 2.2.1 leads to a fast/slow training procedure; and clarify the sequential training protocol and how that prevents drift/maintains accuracy on prior tasks.
-	Is FSA used for both text and image adapters? In ablation experiments where FSA is disabled, are there still 2 sets of adapters or only a single one? A reduced number of parameters could impact performance.
-	Was EMA considered for the fast and slow training procedure?
-	Have authors visualised how parameter gamma evolves over time ? How can this parameter be interpreted in the context of fast and slow learning?

---

### Official Review · Reviewer_MJBv · 2025-11-01

**Soundness:** 2
**Presentation:** 3
**Contribution:** 2
**Rating:** 2
**Confidence:** 3

**Summary:**

This paper studies a continual learning problem with imbalanced task data (TICL). To address the stability-plasticity and head-tail learning dilemma, it first proposes a fast-slow adaptation model which learns two adapters (branches) with different weighting for fast and slow knowledge; then proposes a decoupling strategy that first fine-tunes the text encoder and then fine-tunes the image encoder.  The proposed method is evaluated under TICL benchmarks compared with many existing CL methods, including both ViT-based and CLIP-based methods. Results show that the proposed method achieves better performance than baselines.

**Strengths:**

1. This paper studies a practical CL problem with imbalanced task data, which has wide applications in real world scenarios.

2. This paper conducts thorough experiments compared to SOTA CL methods, with different task-arrival orders. Experimental results show the efficacy of the proposed method.

3. The visualization of the paper is good.

**Weaknesses:**

1. Although motivated by the dual-process theory, it is unclear to me why the proposed fast-slow adapter achieves the fast and slow learning.
* In joint learning, Eq. 4 can be written as:
$ FSA(h)= h+\gamma \alpha W\_\{up\}^\{fast\} \sigma (W^\{fast\}_\{down\}h) + (1-\gamma) \beta W\_\{up\}^\{slow\} \sigma ( W^\{slow\}\_\{down\} h) $. Since $\gamma$ is learned as well, how different are $\gamma\alpha$ and $(1-\gamma)\beta$? Will the initialization of MLP in Eq. 3 influence the fast/slow learning?
* How does ‘the fast adapter enables rapid adaptation to new tasks while leaving the base encoder largely unchanged’ (line 180)? An ablation study of what is learned differently by the fast and slow adapter could be helpful.

2. My main concern lies in the analysis of the decoupled text/image encoder learning. Some claims in the analysis lack sufficient theoretical or empirical justification.
* For example, in line 249-252, how to define ‘a stable semantic anchor’? How do the updates move features to the anchor? In Theorem 1, what are the ‘standard optimization settings’? In Fig 3, fine-tuning image encoder first generally performs better than fine-tuning text encoder first in the descending order, which seems against the claim that first fine-tuning text encoder is helpful.

3. In the ablation study Figure 4, the performance difference under different settings is marginal. It could be helpful to show the statistical significance of the scores as well.

**Questions:**

In Eq. 2, should it be $\sigma(W^{fast}\_{down}h)$ instead of $f(W^{fast}\_{down}h)$?

---

### Note · Authors · 2025-11-12

I have read and agree with the venue's withdrawal policy on behalf of myself and my co-authors.